# D2T2: Decision Transformer with Temporal Difference via Steering Guidance

## Abstract

Despite the promising performance of Decision Transformers (DT) on a wide range of tasks, recent studies have found that the performance of DT may largely be dependent on the specific characteristics of the task of interest including, most importantly, the stochasticity of the environment. We first focus on this issue and prove that a well-trained DT can recover the optimal trajectory almost surely in an environment with random initial states but deterministic transition and rewards, explaining the remarkable performance of DT in deterministic tasks. Notably, it follows from our analysis that for stochastic transition and rewards, the performance of DT may degrade significantly due to the growing variance of returns-to-go (RTG) accumulated over the horizon. To this end, we extend DT to *Decision Transformer with Temporal Difference via Next-State Guidance* (D2T2) which addresses the growing variance problem of RTGs and leads to significantly improved performance in stochastic tasks. D2T2 maps the current state to a guiding vector that steers DT toward high-reward regions where the expected returns are approximated by temporal difference learning. This approach also addresses another severe challenge faced by DT which is its requirement of RTGs as input upon evaluation/deployment. Experimental results on various stochastic tasks and D4RL environments are provided to establish the superior performance of our proposed method compared to the state-of-the-art (SOTA) offline reinforcement learning methods.

## 1 Introduction

One of the most fundamental reasons that reinforcement learning (RL) and supervised learning have been considered as distinct learning problems is that, unlike the scenarios in supervised learning, there is no straightforward label that classifies actions as "good" versus "bad" in the settings of RL. A recently proposed framework Reinforcement learning Via Supervised learning (RvS) includes a variety of methods that attempts to use supervised learning to solve RL tasks (Ding et al., 2019; Ghosh et al., 2021; Codevilla et al., 2018; Neumann & Jan Peters, 2008; Peng et al., 2019). RvS methods have shown promising performance in offline RL tasks, outperforming conventional RL methods such as Temporal-Difference (TD) learning (Emmons et al., 2021). In particular, Decision Transformer (DT) (Chen et al., 2021) has received significant attention. DT recasts RL problems into sequence prediction problems where at each time step DT consumes the realized trajectory so far and generates an action to take. One of the most pioneering contributions of DT is that when taking input DT replaces the instantaneous rewards at all previous time steps with a signal called returns-to-go (RTG) which represents the cumulative reward that the agent should achieve in future. This innovation has led to great empirical success. However, the limitations of DT are also investigated from different perspectives, including the stochasticity issue (Paster et al., 2022; Villaflor et al., 2022), the stitching ability (Yamagata et al., 2023), and return specification (Correia & Alexandre, 2022). Among all the factors that can effect the performance of DT, the stochasticity of environment has received significant attention. Specifically, it has been observed that the performance of DT in stochastic environments is much less competitive than that in deterministic ones (Paster et al., 2022).

In this work, we first formalize the learning problem faced by DT and theoretically prove that DT can recover the optimal trajectory almost surely in an environment with random initial state but deterministic transition and reward. Notably, our analysis reveals that the performance of DT can be significantly degraded due to the growing variance of RTG that has been accumulated over the horizon in a stochastic environment. To verify the correctness of our analysis, we implement a

variation of DT where RTG is replaced by the output of an approximated optimal value function and empirically verify that it outperforms the original DT. This approach utilizes the power of TD learning to overcome the variance problem, reminiscent of the highly effective collaboration between policy optimization and TD learning in Actor-Critic algorithms (Konda & Tsitsiklis, 1999). Moreover, we observe that the learning problem faced by the original DT involves long horizon prediction problems which are challenging to conquer. This inspires us to transform the original learning problem into one that involves shorter horizon thus easier to learn.

Motivated by our analysis and observation, we propose *Decision Transformer with Temporal Difference via Next-State Guidance* (D2T2) that integrates DT with approximated TD learning. Specifically, D2T2 maps the current state to a guiding vector that steers DT toward high-reward regions where the expected returns are approximated by TD learning. Through TD learning, D2T2 addresses the variance problem of RTG and demonstrates significantly improved performance in stochastic tasks compared to DT. With a transformed learning problem (see Section 4) that involves shorter horizon, D2T2 further improves the already competitive performance of DT in deterministic tasks. As a bonus strength, D2T2 does not require RTG as input and thus eliminates the issue of manually deciding the RTG during evaluation, which is one of the most severe challenges faced by the original DT. We benchmark our proposed method on 2 illustrative stochastic examples (*i.e.,* FrozenLake and Tailgate driving (Villaflor et al., 2022)), 2 stochastic CARLA benchmarks (Dosovitskiy et al., 2017), and 3 suites (18 tasks) from D4RL (Gym-MuJoCo, AntMaze, and FrankaKitchen) (Fu et al., 2020). This set of environments with tasks of different levels of difficulty enable us to comprehensively investigate the capability of our approach. Our method demonstrates a competitive performance compared with state-of-the-art (SOTA) baselines, including both return-conditioned and goal-conditioned methods.

## 2 MOTIVATIONS TOWARDS AN IMPROVED DECISION TRANSFORMER (DT)

### 2.1 PROBLEM SETUP FOR DT

Sequential decision-making problems can be formulated as Markov Decision Processes (MDPs), defined by a tuple $(\mathcal{S}, \mathcal{A}, \mathcal{R}, P, T)$, where $\mathcal{S}$ and $\mathcal{A}$ are the set of states and actions respectively; $\mathcal{R}$ is the reward function where $r_t = \mathcal{R}(s_t, a_t)$ is the instantaneous received reward at time step $t$ by taking action $a_t$ in state $s_t$; $P(s'|s, a)$ is the transition probability for $s, s' \in \mathcal{S}$, $a \in \mathcal{A}$; and $T$ is the horizon. In this section, we consider finite-horizon MDPs without discounting factor, a scenario that DT focuses on. The ultimate goal is to learn an optimal policy $\pi^*$ that maximizes the expected total return, *i.e.,*

$$\pi^* \in \arg\max_{\pi} \left\{ \mathbb{E}_{\rho_\pi} \left[ \sum\nolimits_{t=0}^{T} \gamma^t r_t \right] \right\}, \tag{1}$$

where $\rho^\pi$ is the state-action distribution under policy $\pi$ and $r_t$ is a random variable that represents the instantaneous reward received by the agent at time $t$. In the offline setting, the optimal policy $\pi^*$ is learned with a fixed set of trajectories $\mathcal{D}$ storing the transitions $(s, a, r, s')$. We note that $\mathcal{D}$ is usually collected over some behavior policy $\pi_b$ which can be either a single policy or a mixture of policies and they are considered unknown.

### 2.2 THE SUCCESS OF DT IN DETERMINISTIC ENVIRONMENTS

At each time step $t$, DT takes as input the realized trajectory and a signal called returns-to-go (RTG) $R_t$, and generates an action $a_t$ to take. The success of DT relies on RTG, which is defined as

$$R_t = \sum_{k=t}^{T} r_k, \tag{2}$$

*i.e.,* the cumulative return obtained from time step $t$ to the end of the trajectory. We note that in general $R_t$ is not accessible at time step $t$ but for an offline trajectory from $\mathcal{D}$, $R_t$ can be computed from all the subsequent rewards received after time step $t$. DT transforms RL problems into supervised learning problems, *i.e.,* predicting a label for a given input feature. A common interpretation of the prediction problem faced by DT is the following:

[Prediction Problem for DT] *Given the previous states and actions, what is the action that can most likely generate the cumulative reward of the amount $R_t$?*

In other words, let $\tau_t$ denote the realized trajectory until time step $t$ and $f^*$ denote a well-trained (optimal) DT, then

$$f^*(\tau_t, t, R_t) = \arg\max_a \left\{ \mathbb{P}\left( \sum_{k=t}^{T} r_k = R_t | \tau_t, a_t \right) \right\}. \tag{3}$$

Suppose we have successfully trained the model for this prediction problem. Then if we are aware of the maximal cumulative rewards or have some desired reward $R^*$, we can input $R_t = R^*$ at time step $t = 0$ and at each time step $t > 0$, we set $R_t = R_{t-1} - r_{t-1}$, *i.e.,* we decrease $R_{t-1}$ by the received rewards $r_{t-1}$ in last time step, and then input $R_t$ into the model for prediction of $a_t$ that is the most likely action to return us the cumulative reward of amount $R_t$. If $R_t$ is the true maximum possible reward for all $t \geq 0$ and DT is well-trained so that the generated $a_t$ is the action that can realize the maximum possible reward with highest probability for at all $t \geq 0$, then the sequence of predicted actions $\{a_i\}_{i=0}^{T-1}$ should be the sequence of actions that maximizes the probability of obtaining $R^*$.

While the above argument is intuitively appealing, we next show rigorously that when DT is well-trained and the environment is deterministic, then the DT is guaranteed to return the optimal sequence of actions that return the maximum cumulative reward. Moreover, it provides insights for understanding the less competitive performance of DT in stochastic environments.

**Proposition 1** (Decision Transformer gives the correct signal almost surely in deterministic environment). *Assume that the environment is deterministic, i.e, both the state transition and the reward function are deterministic given the current state and action. Suppose that we are maximizing the cumulative reward in a trajectory with horizon $H$. For any initial state $s_0$, let the optimal trajectory be denoted as $\{s_0(s_0^*), a_0^*, r_0^*, \ldots, s_H^*, a_H^*, r_H^*\}$ such that for all $t > 0$, $s_t^* = p(s_{t-1}^*, a_{t-1}^*)$ and $r_t^* = r(s_{t-1}^*, a_{t-1}^*)$ where $p$ and $r$ are respectively the deterministic transition and reward functions. For presentation simplicity we assume that the optimal trajectory is unique. Let $R_t^* = \sum_{t'=t}^{H} r_{t'}^*$ be the future maximum cumulative reward that can be collected given the previous trajectory at time step $t$.*

*Then we have the following three facts for any $s_0 \in S$:*

1. *At $t = 0$, a well-trained Decision Transformer $f^*$ will output $a_0^*$ if the input $R_t = R_0^*$. (If the optimal trajectory is not unique, then the outputted $a_0^*$ must be the first action of some optimal trajectory).*

2. *At $t > 0$, if the input to DT is $R_t^*$, then the output action must be $a_t^*$.*

3. *At $t > 0$, the recursively computed signal $R_t = R_{t-1} - r_t$ must equal to $R_t^*$.*

*Combining the above facts, we conclude that DT outputs the optimal trajectory almost surely.*

*Proof of Theorem 1.* Please see in Appendix. □

The above theorem explains the power of DT in deterministic environment. The key takeaway for the above theorem is that *the success of DT relies on the successful estimation of $R_t^*$*. And in deterministic environment, $R_t^*$ can be estimated almost surely correct if the initial signal is correct. However, in stochastic environment, every recursive step of computation for $R_t$ will accumulate variance. This will result in a signal with expanding variance over the horizon.

## 2.3 IMPROVED SIGNAL FOR DT IN STOCHASTIC ENVIRONMENTS

An intuitive solution to reduce the variance is to use the Markov property of the environment and to build a signal that only uses $s_t$ at time step $t$. To better motivate this solution, we note that this solution shares the same motivation and theoretical support with the usage of TD learning to approximate the cumulative future rewards in the Actor-Critic method (Konda & Tsitsiklis, 1999) where the variance of the Monte-Carlo estimates of cumulative rewards causes unstable gradient estimation. Hence, we would like a signal that tells us the maximum amount of future cumulative reward that any agent can collect, using only the information in the current state $s_t$. This notion is exactly the optimal value function, *i.e.,* the value function of the optimal policy.

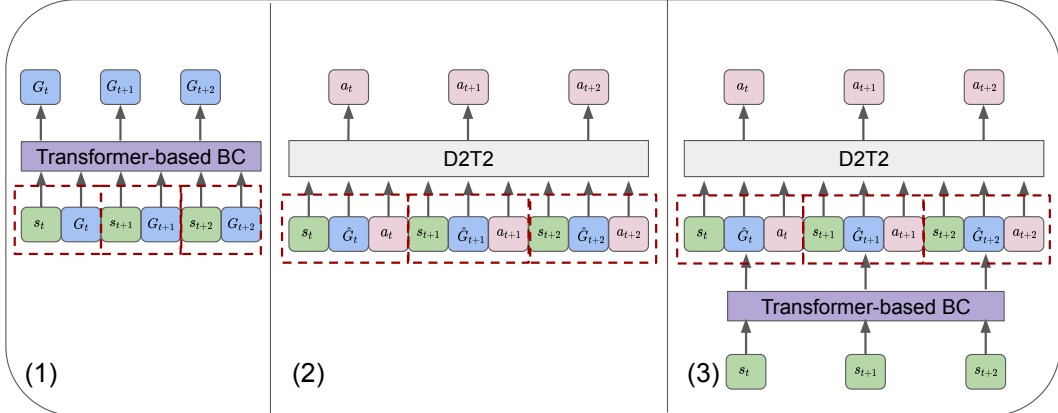

Figure 1: Overview of the D2T2 Framework. (1) D2T2 first extrapolates a steering guidance function with transformer-based behavior cloning where the labels for cloning are inferred from the offline dataset through TD learning. (2) In the training stage, D2T2 aligns the generated steering-guided action with the observed action via supervised learning. (3) During evaluation, D2T2 employs the learned steering guidance function to generate actions that lead to optimal return.

To verify the correctness of this conjecture, we implement a modification of DT where we replace RTG with the output of the value function from a well-trained Q-learning model and show that it successfully outperforms the original DT in stochastic tasks (please refer to Section 4 for details). This demonstrates that TD learning can address the problem of growing variance of input signal to DT in stochastic environments.

While TD learning can address the challenge of growing variance, the fundamental prediction problem faced by DT remains: to predict the action that can most likely lead to a desired amount of cumulative reward. We observe that this is a challenging long horizon prediction problem because it involves consideration of *all* the subsequent trajectory as the cumulative future reward is the sum of all the instantaneous rewards collected in the remaining time steps of the trajectory. To reduce the horizon in consideration, we propose a new and less challenging prediction problem for DT:

[Modified Prediction Problem for DT] *What is the action that can most likely lead to a desired state?*

The proposed prediction problem does not necessarily involve the complete horizon from the current time step to the end because DT can step into the desired state at any future time step, thus decreasing the inherent difficulty of its prediction problem.

With all the aforementioned analysis and motivations, we propose *Decision Transformer with Temporal Difference via Steering Guidance* (D2T2) which has an input signal that can modify the original prediction problem of DT into the less challenging one above. Notably, the input signal employs TD learning to address the growing variance problem and eliminates the need of RTG upon evaluation which is one of the most severe challenges faced by the original DT.

## 3 D2T2: DT WITH TEMPORAL DIFFERENCE VIA STEERING GUIDANCE

D2T2 converts the tasks of decision-making to sequence modeling problems via supervised learning on an offline dataset $\mathcal{D}$ to extrapolate a guidance-steered policy (GSP) $\pi_\theta(a_t|s_{t-k:t}, a_{t-k:t-1}, \widehat{G}_{t-k:t})$ , *i.e.*, the action $a_t$ at time step $t$ is determined not only based on the sequence of prior states $s_{t-k:t}$ and actions $a_{t-k:t-1}$ but also a sequence of steering guidance signals (SGs), $\widehat{G}_{t-k:t}$. At each time step $t$, $\widehat{G}_t$ provides a signal that can lead $\pi_\theta$ towards achieving a pre-determined goal. In the original DT, $\hat{G}_t$ at time step $t$ is $R_t$ (i.e., the RTG at time step $t$). Distinct to DT, D2T2 employs a novel SG that involves a distinct goal with shorter horizon and utilize TD learning to address the growing variance problem of RTG. This leads to superior performance of D2T2 over the original DT, especially in stochastic tasks (see Section 4). For clarity of notation, in the following sections, we use $\widehat{G}_t$ to denote the signal input into D2T2 that is generated by a learned parametric function $\bar{g}_\zeta$, and $G_t$ to denote a

---

**Algorithm 1** SG Learning for D2T2

---

1: **Input:** Training dataset $\mathcal{D} = \{\tau_1, \tau_2, ..., \tau_n\}$ of training trajectories, approximated optimal value function $V(s)$ from TD learning
2: **Output:** Learned function for SG generation $\bar{g}_\zeta(s_{t-k:t})$
  {**Step I** in Section 3.1 }
3: **for** each $\tau_i = \{s_0, a_0, s_1, a_1, ...\}$ in $\mathcal{D}$ **do**
4:    **for** $t = 0$ **to** $T - 1$ **do**
5:       $G_t = g(s_t) = \arg\max_{s_j}\{\gamma^{j-t}V(s_j)|j > t, s_j \in \tau_i\}$
6:       Add $G_t$ to $\tau_i$
7:    **end for**
8:    $G_t = s_T$
9:    Add $G_t$ to $\tau_i$
10: **end for**
  {**Step II** in Section 3.1 }
11: Behavior cloning with causal transformer by optimizing :

$$\min_\zeta \frac{1}{N}\sum_{\tau_i \in \mathcal{D}}\frac{1}{T}\sum_{s_t \in \tau_i}||g(s_t) - \bar{g}_\zeta(s_{t-k:t})||^2$$

   with optional VAE.

---

signal that is computed from the information in the offline dataset $\mathcal{D}$ and will be used in the training of $\bar{g}_\zeta$. We next introduce the SG used by D2T2.

### 3.1    LEARNABLE STEERING GUIDANCE (SG) OVER LATENT REPRESENTATIONS

As discussed in Section 2.3, value functions $V(\cdot)$ can guide DT toward achieving the optimal cumulative rewards. However, it requires $V(\cdot)$ to be near-optimal in order to provide effective guidance, which is considered challenging in offline RL due to the limited coverage of the state-action space provided by the offline dataset $\mathcal{D}$. To this end, we leverage variational inference (Kingma & Welling, 2013) to encode guidance provided by the sub-optimal value functions into a compact and expressive latent space, which distills the knowledge acquired from state values as well as environmental transitions and rewards, to formulate the final SGs. The detailed steps for generating SGs are introduced below and summarized in Algorithm 1.

**Step I.** As discussed in Section 2.3, compared to the amount of desired future cumulative rewards, a desired next state can potentially decrease the horizon of the decision problem, thus improving the learning efficiency. Accordingly, the first step toward constructing the SGs is to generate a mapping function $g : \mathcal{S} \rightarrow \mathcal{S}$ such that each state $s_t \in \tau_i$ is mapped to a corresponding desired next state $G_t = g(s_t) \in \tau_i$ that has the maximum value, *i.e.*,

$$G_t = g(s_t) = \arg\max_{s_j}\{\gamma^{j-t}V(s_j)|j > t, s_j \in \tau_i\}, \tag{4}$$

where $V(\cdot)$ is an approximated optimal value function that estimates the maximum expected cumulative return from state $s$, *i.e.*, $V(s) \approx \mathbb{E}_{\rho^{\pi^*}}[\sum_{k=0}^{T}\gamma^k r_{t+k+1}|s_t = s]$ where $\pi^*$ is the optimal policy. In practice, though one can use any temporal difference (TD) algorithms for optimal value function approximation, the implicit Q-learning (IQL) Kostrikov et al. (2022) remains a compelling algorithm for learning $V(\cdot)$, so we alternatively use Q-learning for descriptions of our methods and experiments. We note the purpose of the discount factor $\gamma$ in equation 4 is to motivate early achievement of desired next state. In other words, if two future states have the same value, then the one that appears earlier in the trajectory is more desired as this can help improve the future cumulative reward.

**Step II.** As discussed in Section 1, a drawback of the original DT is that the RTGs are not available during evaluation/deployment after the training is completed, as they depend on future information. Hence, they become hyper-parameters to be tuned which may lead to unstable performance. The guidance provided from the value function, following equation 4, faces the same limitation as it depends on the value of future states. To address this, we employ behavior cloning with causal transformer Chen et al. (2021), as demonstrated in Figure 1, to learn a function $\bar{g}_\zeta : \mathcal{S}^k \rightarrow \mathcal{S}$ that

extrapolates the function $g(\cdot)$ by minimizing the squared loss, *i.e.*,

$$\min_{\zeta} \frac{1}{N} \sum_{\tau_i \in \mathcal{D}} \frac{1}{T} \sum_{s_t \in \tau_i} ||g(s_t) - \bar{g}_\zeta(s_{t-k:t})||^2, \quad (5)$$

where $\zeta$ is the parameters of the causal transformer, $N$ is the number of trajectories in the offline dataset $\mathcal{D}$, and $T$ is the horizon of the environment. Consequently, at each time step $t$, given the sequence of prior states $s_{t-k:t}$, the SG $G_t = g_t(s_t) \approx \bar{g}_\zeta(s_{t-k:t})$ can be obtained without leveraging any information from the future. We choose a causal transformer over other architectures (*e.g.*, MLP, RNN, LSTM, etc.) to ensure the model has access to the whole long-horizon sequence. Here we specifically choose the transformer architecture as they have shown to be effective in processing long input sequences (Melnychuk et al., 2022).

Moreover, given that the approximated optimal value function $V(\cdot)$ is highly likely to be sub-optimal if the offline dataset does not comprehensively cover the state and action space (Levine et al., 2020), its approximation errors can be propagated into $\bar{g}_\zeta(\cdot)$ whose training requires $V(\cdot)$ as shown by equation 5 and equation 4. As a result, directly using $\bar{g}_\zeta(\cdot)$ as the SG could be problematic, as the errors from the previous two steps can both be propagated into DT, in addition to the supervised learning error from the training itself. To resolve this, we leverage variational inference (Kingma & Welling, 2013) to concentrate the learned knowledge into a compact and expressive latent space via a variational auto-encoder (VAE), leading to $\widehat{G}_t \sim q_\psi(\cdot|\bar{g}_\zeta)$, where $q_\psi(\cdot|\bar{g}_\zeta)$ is the approximated posterior that encodes $\bar{g}_\zeta$ to the latent space. In practice, we observe that variational inference is not always necessary. For example, for complex tasks with complicated environment dynamic or high-dimensional states, VAE can benefit the performance. On the other hand, for simple tasks, the negative impact induced by the learning error of VAE may outweigh the benefits brought by the latent space. In this case, variational inference should be employed with caution. Hence, the SG inputs of D2T2 in Figure 1 can be either $\widehat{G}_t = \bar{g}_\zeta(s_{t-k:t})$ or $\widehat{G}_t \sim q_\psi(\cdot|\bar{g}_\zeta)$, depending on whether or not a latent representation is employed. Due to the space limit, please refer to Appendix B for additional details. In our experiments in Section 4, we explicitly report which SG is used by D2T2.

### 3.2 D2T2 Training & Evaluation

In contrast to the original DT where the RTGs are not available upon evaluation/deployment, the SG of D2T2 can be determined without the need of future information following the design above. Specifically, at step $t$, the inputs to D2T2 can be formulated as

$$\tau_{input} = (s_0, a_0, \widehat{G}_0..., s_{t-1}, a_{t-1}, \widehat{G}_{t-1}, s_t, \widehat{G}_t), \quad (6)$$

with $\widehat{G}_t \sim q_\psi(\cdot|\bar{g}_\zeta(s_{t-k:t}))$ following the step above where $\bar{g}(s_{t-k:t})$ is deterministic as introduced above. Consequently, D2T2 can be trained with only the offline trajectories $\tau_i \sim \mathcal{D}$ to minimize the squared loss between the actions provided in the offline trajectory, $a \sim \tau_i$, and the actions predicted from D2T2, $a_t = \pi_\theta(s_{t-k:t}, a_{t-k:t-1}, \widehat{G}_{t-k:t})$, following the regular supervised learning schema as in the original DT Chen et al. (2021). Upon evaluation, D2T2 takes as the input the sequence equation 6 (up to the current step $t$), and subsequently generate the action, following $a_t = \pi_\theta(s_{t-k:t}, a_{t-k:t-1}, \widehat{G}_{t-k:t})$. The training and testing stages are summarized in Algorithm 2.

## 4 Experiments

In this section, we provide a comprehensive empirical study of our method and compare it with other state-of-the-art (SOTA) methods. We consider a broad range of tasks, including 2 illustrative stochastic examples (*i.e.,* Tailgate driving and FrozenLake), 2 stochastic CARLA benchmarks, and 3 suites from D4RL (Gym-MuJoCo, AntMaze, and FrankaKitchen) (Fu et al., 2020). First we corroborate the correctness of our analysis in Section 2.2 that motivates the integration of DT with the value function. Specifically, we implement VDT which is a variation of the original DT that directly replaces RTG with an approximated optimal function. After verifying the correctness of our theoretical insights, the remaining experiments focus on demonstrating the competitive performance of our proposed algorithm D2T2.

In the stochastic tasks, we mainly compare methods that follow a similar architecture branch: Decision Transformer (DT) Chen et al. (2021), Trajectory Transformer (TT) Janner et al. (2021), SeParated

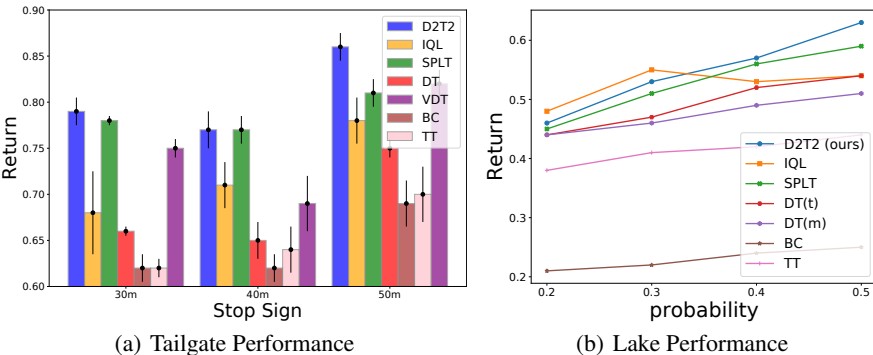

(a) Tailgate Performance

(b) Lake Performance

Figure 2: (a) Tailgate task performance with different stop signs (data optimality): the smaller the stop sign is, the higher the crash rate of the data is. (b) FrozenLake task performance with different probability (stochasticity): the smaller the x-value is, the larger the stochasticity is.

Latent Trajectory Transformer (SPLT) Villaflor et al. (2022), and transformer-based behavior cloning (BC) Chen et al. (2021). We also include a strong offline Q-learning (IQL) for comparison. For D4RL tasks, we compare methods that are representative of both Q-learning and RvS. In the former category, we compare Conservative Q-learning (CQL) Kumar et al. (2020) and IQL. In the latter category, we consider both (1) fully-connected architectures: RvS-R and RvS-G (learning with either conditioned on goals or rewards Emmons et al. (2021) and (2) transformer-based models: DT, TT, SPLT, and transformer-based behavior cloning (BC). In addition, we also include mildly conservative Q-learning (MCQ) (Lyu et al., 2022) to Gym-MuJoCo suite and contrastive RL (CL) (Eysenbach et al., 2022) to AntMaze-v2 suite as strong baselines. We note that D2T2 employs VAE in all the tasks other than the two illustrative tasks (Tailgate and FrozenLake) due to their simplicity. Due to space limitation, please refer to Appendix C for detailed experimental setup, ablation studies, and additional experimental results.

**Tailgate Driving.** We conduct experiments on the Tailgate driving task to verify our theoretical insights that motivate the integration of TD with DT. The goal of the tailgate driving problem is to maximize the ego vehicle's distance within one episode following a leading vehicle in the same direction on a 1-dimension path. We collect trajectories following the prior work Villaflor et al. (2022) for an autonomous driving problem whose state space is the absolute position and velocity of both the ego and leading vehicles, $s \in [e_x, e_v, l_x, l_v]$, and the action space is the acceleration of the ego vehicle $a \in [-1, 1]$. The leading vehicle will either begin hard-braking at the last possible moment to stop just before the $d$ mark meters and then resume going forward or speed up to the maximum speed and continue for the entire trajectory. Therefore, this scenario is a stochastic event to the ego vehicle. We collect around 100k samples for each dataset with different $d$ as the stop sign and its corresponding crash rate as the data quality, which is shown in Table 4 in Appendix. The reward for each time step records the normalized distance $d_n = \frac{d}{d_{max}}$, where $d$ is the actual distance the ego vehicle moving forward and $d_{max}$ is the maximum moving distance between the ego and leading vehicles considering the safe following distance. We report the task performance with standard error in Figure 2(a). Our empirical results show that VDT improves DT substantially, which justifies our proof and intuition in Section 2.3. Note that DT here has already been hand-tuned. The DT conditioned on the maximum return in the dataset performs worst, which is omitted. Our empirical results in this stochastic problem show that D2T2 has a significantly higher return compared with the original DT and VDT, which supports our previous argument. Therefore, we focus on D2T2 as our method for the remaining experiments.

**Stochastic Frozenlake.** Frozenlake environment with stochastic transitions is visualized in Figure 3 in Appendix. We conduct experiments on the datasets with different probabilities $p$ of moving toward the intended direction and the probability $1 - p$ of moving to either between two sides of the intended directions to evaluate with different levels of stochasticity. The lower is $p$, the more stochastic is the environment. We vary $p$ value for data collection and evaluation. The offline trajectories have length of 100 and are acquired via training a DQN Mnih et al. (2013) policy in gym environment. In Figure 2(b), we show that our D2T2 earns higher returns among all stochasticity levels compared with most of the baselines, including SPLT which is state-of-the-art in solving stochasticity issues of

Table 1: Success rate (%) and Speed (m/s) on NoCrash benchmark. We evaluate D2T2 with baselines for 10 seeds through all 25 routes in the unseen Town02. DT(m) is DT conditioned on the maximum return in the dataset. DT(t) is DT with a hand-tuned conditional return. For both Success (%) and Speed (m/s) a larger value is better.

| Metric | BC | TT | DT(m) | DT(t) | IQL | SPLT | D2T2 |
|---|---|---|---|---|---|---|---|
| **Success (%)** | 92.2±0.5 | 85.4±2.5 | 89.7±6.2 | 94.2±2.9 | 97.5±0.4 | 95.1±2.6 | **98.3±0.8** |
| **Speed (m/s)** | 2.44±0.01 | 2.62±0.3 | 2.70±0.06 | 2.76±0.03 | 2.79±0.06 | 2.75±0.09 | **2.81±0.11** |

Table 2: Multiple metrics on Leaderboard benchmark. We evaluate D2T2 with baselines for 10 seeds and on Leaderboard devtest routes. DT(m) is DT conditioned on the maximum return in the dataset. DT(t) is DT with a hand-tuned conditional return. For both Total Score and Completion (%) a larger value is better, while for Collision (/km) and Infraction (/km) a smaller value is better. A larger performance value is better.

| Metric | BC | TT | DT(m) | DT(t) | IQL | SPLT | D2T2 |
|---|---|---|---|---|---|---|---|
| **Total score** | 53.4±7.1 | 56.2±8.5 | 62.3±11.3 | 68.6±4.5 | 63.6.5±6.8 | 65.8±4.3 | **70.2±4.5** |
| **Completion (%)** | 94.2±4.5 | 70.6±10.2 | 95.6±4.8 | 98.3±2.7 | **100.0±0.0** | 93.5±8.6 | **100.0±0.0** |
| **Collision (/km)** | 4.1±1.2 | 2.8±2.2 | 1.8±1.7 | **1.7±0.5** | 2.3±0.5 | 2.5±0.3 | 1.8±1.3 |
| **Infraction (/km)** | 2.4±2.2 | **0.0±0.0** | 2.6±0.9 | 2.5±0.8 | 2.2±0.7 | 2.2±1.3 | 2.1±0.8 |

DT. DT(m) is DT conditioned on the maximum return in the dataset. DT(t) is DT with a hand-tuned conditional return. The standard error is small enough for all methods to be ignored. We observe that IQL is a strong comparison for really low $p$ while our performance is still competitive. In addition, when $p$ increases to 0.6, which is still not a large probability, D2T2 outperforms IQL significantly.

**CARLA NoCrash Benchmark.** We evaluate our method on the CARLA NoCrash (Dosovitskiy et al., 2017; Codevilla et al., 2019) benchmark whose goal is to navigate in a suburban town to a desired goal waypoint from a predetermined start waypoint without crash, considering safety as a priority Wen et al. (2020); Hsu et al. (2022) in tasks such as autonomous driving. The benchmark consists of 2 towns *Town01* and *Town02*, each with 25 different routes. We train our method D2T2 as well as all the baselines on the Town01 dataset and then evaluate them in the unseen Town02 routes. In Table 1, we present 2 metrics for comparison, observing that DT(t) with tuning target return RTG does improve success rate and speed against DT(m) with maximum return in the dataset. However, it would be difficult to estimate the best target return without online evaluation or prior domain knowledge, especially since the training and testing are in different scenarios, *i.e.,* Town01 and Town02. By contrast, D2T2 leverages the benefit of Q-learning with a properly designed guidance signal, leading to both the highest success rate and speed and outperforming IQL.

**CARLA Leaderboard Benchmark.** Next, we evaluate our method on a modified version of the CARLA Leaderboard benchmark following the setting in Villaflor et al. (2022). Compared with NoCrash Benchmark, this task involves more maneuvers like lane-changing in urban and highway situations as well as 8 additional variables. The results are summarized in Table 2. Total scores are calculated with route completion rate, collisions per kilometer (/km), and infractions (/km), which is the main indicator of the performance for all methods. Among all the baselines, we would like to emphasize the high performance in DT(t) is similar to the NoCrash benchmark in that manually tuning the target return is arbitrary. Although SPLT is able to disentangle the world dynamics and the agent policy, resulting in a competitive performance in such complexity of CARLA Leaderboard scenarios, our better total score indicates that steering guidance is a more proper condition variable versus RTG, incorporating value function.

**D4RL Suites.** As discussed in Section 2.2, DT has competitive performance in deterministic or near-deterministic tasks, such as the ones in D4RL suites. From Table 3, we observe that D2T2 outperforms the original DT with a competitive performance compared to strong offline Q-learning methods (*e.g.,* IQL and CQL) in MuJoCo tasks. We report the other complete results with standard error for AntMaze and FrankaKitchen tasks in Table 11, and 12 respectively. Specifically, D2T2 outperforms DT in AntMaze environments significantly in both Table 11 and Figure 4 in Appendix. We believe that D2T2 indirectly improves its stitching ability (*i.e.,* learning the optimal policy from sub-optimal trajectories) by integrating with Q-learning Yamagata et al. (2023).

Table 3: Averaged normalized scores on Gym-MuJoCo suite. We use the results reported from the TT paper for BC, DT, and TT and the results from the IQL paper for CQL and IQL. For RvS-R, SPLT, and MCQ, we use the results reported from their own papers respectively. We report the mean and standard error for our method over 10 seeds. The top scores are bolded.

| Environment | BC | RvS-R | DT | TT | SPLT | CQL | IQL | MCQ | D2T2 |
|---|---|---|---|---|---|---|---|---|---|
| halfcheetah-Med-Expert-v2 | 59.9 | 92.2 | 86.8±1.3 | **95.0±0.2** | 91.8±0.5 | 91.6 | 86.7 | 87.5±1.3 | 90.9±0.8 |
| walker2d-Med-Expert-v2 | 36.6 | 106.0 | 108.1±0.2 | 101.9±6.8 | 108.6±1.1 | 108.8 | 109.6 | 114.2±0.7 | **109.9±1.7** |
| hopper-Med-Expert-v2 | 79.6 | 101.7 | 107.6±1.8 | 110.0±2.7 | 104.8±1.1 | 105.4 | 91.5 | 111.2±0.1 | **114.8±0.5** |
| **average-Med-Expert-v2** | 58.7 | 100.0 | 100.8 | 102.3 | 100.7 | 95.9 | 101.9 | 104.3 | **107.1** |
| halfcheetah-Med-Replay-v2 | 4.3 | 38.0 | 36.6±0.8 | 41.9±2.5 | 42.7±0.3 | 45.5 | 44.2 | 56.8±0.6 | **62.5±0.2** |
| walker2d-Med-Replay-v2 | 36.9 | 60.5 | 66.6±3.0 | 82.6±6.9 | 57.7±4.7 | 77.2 | 73.9 | 91.3±5.7 | **101.8±5.2** |
| hopper-Med-Replay-v2 | 27.6 | 73.5 | 82.7±7.0 | 91.5±3.6 | 75.0±23.8 | 95.0 | 94.7 | **101.6±0.8** | 92.8±4.7 |
| **average-Med-Replay-v2** | 22.9 | 57.3 | 62.0 | 72.0 | 58.5 | 72.6 | 70.9 | 83.2 | **85.7** |
| halfcheetah-Medium-v2 | 43.1 | 41.6 | 42.6 ±0.1 | 46.9±0.4 | 44.3±0.7 | 44.0 | 47.4 | 64.3±0.2 | **79.6±0.8** |
| walker2d-Medium-v2 | 77.3 | 71.7 | 74.0±1.4 | 79.0±2.8 | 77.9±0.3 | 72.5 | 78.3 | **91.0±0.4** | 89.2±0.4 |
| hopper-Medium-v2 | 63.9 | 60.2 | 67.6±1.0 | 61.1±3.6 | 53.4±6.5 | 58.5 | 66.3 | **78.4±4.3** | 74.8±3.4 |
| **average-Medium-v2** | 61.4 | 57.8 | 61.4 | 62.3 | 58.5 | 58.3 | 64.0 | 77.9 | **81.2** |
| **average-Gym-v2** | 47.7 | 71.7 | 74.7 | 78.9 | 72.9 | 77.6 | 76.9 | 88.5 | **91.3** |

## 5 RELATED WORKS

Offline RL learns the policy purely from previously collected data without online interaction, which is appealing when real-world exploration with untrained policies is dangerous or costly. The distribution shift in offline RL, coming from the difference between the learned policy and behavior policies, should be mitigated. Prior works have constrained or regularized dynamic programming to mitigate deviations from the behavior policy Kumar et al. (2019); Wu et al. (2019); Fujimoto et al. (2019); Kumar et al. (2020); Fujimoto et al. (2021). On the other hand, Decision Transformer (DT) Chen et al. (2021) and Trajectory Transformer (TT) Janner et al. (2021) leverage transformer architecture to fit a reward-conditioned policy and model trajectory distributions respectively.

The scope of DT has been extended by following works, such as Online DT Zheng et al. (2022), Multi-Game DT Lee et al. (2022), and CDT Liu et al. (2023). In addition, several works target the vulnerability of DT with stochasticity issues. For example, ESPERPaster et al. (2022) learn trajectory representations disentangled from environmental dynamics via adversarial clustering and SPLTVillaflor et al. (2022) learn environmental stochasticity and agent policy separately with 2 transformers. Similarly, DoC Yang et al. (2023) predicts the trajectories' representations by minimizing the mutual information between the representation and the environment transition.

Alternatively, we incorporate DT with Q-learning to alleviate the stochasticity problem for DT. Several works have tried to combine DT with dynamic programming (*e.g.,* Q-learning). Among them, QDT Yamagata et al. (2023) relabels RTG in DT with precomputed conservative value functions to improve DT's stitching ability. EDT Wu et al. (2023) adjusts the context length of DT with interpolation between trajectory stitching and behavior cloning. However, both of them mainly tackle the stitching problem and are not evaluated in stochastic tasks. Another recent work WT Badrinath et al. (2023) aims to utilize behavior cloning to mitigate stitching and instability initialization without considering the learned value functions.

## 6 CONCLUSION

In this work, we investigate the Decision Transformer to understand its strength and weakness. Specifically, our analysis provides an explanation for the strong performance of DT in deterministic task, reveals a potential reason for its less competitive performance in stochastic environment, and suggests two potential improvements. Motivated by these insights, we propose a new approach D2T2 with a TD-learned guiding signal that significantly improves the performance of DT in stochastic tasks. On a variety of environments and tasks, our method has demonstrated SOTA performance, revealing the promising potential of combining DT with Temporal Difference learning.

## REPRODUCIBILITY STATEMENT

We have submitted the code of our implementation of D2T2 as supplementary material. Algorithms and proposed framework are described in Section 3 and B. Information about the benchmarks and hyper-parameters guidance are detailed in Section 4 and C.

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

APPENDIX

## A    PROOF OF THEORETICAL RESULTS

**Theorem 1** (Decision Transformer gives the correct signal almost surely in deterministic environment). *Assume that the environment is deterministic, i.e, both the state transition and the reward function are deterministic given the current state and action. Suppose that we are maximizing the cumulative reward in a trajectory with horizon $H$. For any initial state $s_0$, let the optimal trajectory be denoted as $\{s_0(s_0^*), a_0^*, r_0^*, \ldots, s_H^*, a_H^*, r_H^*\}$ such that for all $t > 0$, $s_t^* = p(s_{t-1}^*, a_{t-1}^*)$ and $r_t^* = r(s_{t-1}^*, a_{t-1}^*)$ where $p$ and $r$ are respectively the deterministic transition and reward functions. For presentation simplicity we assume that the optimal trajectory is unique. Let $R_t^* = \sum_{t'=t}^{H} r_{t'}^*$ be the future maximum cumulative reward that can be collected given the previous trajectory at time step $t$.*

*Then we have the following three facts for any $s_0 \in S$:*

1. *At $t = 0$, a properly trained Decision Transformer will output $a_0^*$ if the input $R_t = R_0^*$. (If the optimal trajectory is not unique, then the outputted $a_0^*$ must be the first action of some optimal trajectory).*

2. *At $t > 0$, if the input to DT is $R_t^*$, then the output action must be $a_t^*$.*

3. *At $t > 0$, the recursively computed signal $R_t = R_{t-1} - r_t$ must equal to $R_t^*$.*

*Combining the above facts, we conclude that DT outputs the optimal trajectory almost surely.*

*Proof of Theorem 1.* Following the proposed conjure that given the input $R_t$, DT is predicting the most likely action $a_t$ that can lead a future cumulative reward of $R_t$, we assume that a well-trained DT can accurately predict such actions at each step.

First, it is important to note that the maximum probability that any action can lead to a future cumulative reward of the amount $R_t^*$ is 1. For any initial state $s_0$, we consider the trajectory with the highest cumulative reward as the optimal trajectory and denote it by $\tau^* \doteq \{s_0(s_0^*), a_0^*, r_0^*, \ldots, s_H^*, a_H^*, r_H^*\}$. If $R_0 = R_0^*$, then $a_0^*$ can lead to a trajectory with a cumulative reward of $R_0^*$ with probability one because the environment is deterministic. If the optimal trajectory is unique, *i.e.,* only $\tau^*$ can has a cumulative reward of $R_0^*$, then because DT is well-trained, it must output $a_0^*$ if the input is $R_0^*$. At time $t > 0$, because the environment is deterministic DT can receive $R_t^*$ if taking action $a_t^*$ and follow the sequence of actions listed in $\tau^*$. This implies that DT can receive a cumulative reward of $R_t^*$ with probability one by taking $a_t^*$. Hence, because DT is well-trained, it will output $a_t^*$.

Next we prove that $R_t = R_t^*$ if $R_0 = R_t^*$. We use induction to prove it. Because we have shown that a well-trained DT will output $a_0^*$ and by assumption that the reward is deterministic, $R_1 = R_0^* - r_0^* = R_1^*$. Hence, the base case is proved. Assume toward induction that at $t = k > 2$, we have $R_{k-1} = R_{k-1}^*$. We have proved that at $t = k$, DT will output $a_k^*$ and due to the fact the reward is deterministic, $r_k = r_k^*$, then by definition of $R_t^*$, $R_k = R_{k-1}^* - r_k^*$. This concludes the proof for the case where the optimal trajectory is unique. For the case where there are multiple optimal trajectories all with cumulative rewards of amount $R^*$, note that the sequence of actions generated by DT must lead to the realization of one of such optimal trajectories. Thereby the proof is concluded.

$\square$

Table 4: Tailgate Driving Environment: information for the autonomous driving tasks, including the number of offline trajectories and crash rate.

| $d$ **mark (m)** | **30** | **40** | **50** |
|---|---|---|---|
| # trajectories | 2728 | 2771 | 2954 |
| crash rate | 22.69% | 17.65% | 10.47% |

---

**Algorithm 2** Training and Evaluation of D2T2

---

1: **Input:** Training dataset $\mathcal{D} = \{\tau_1, \tau_2, ..., \tau_n\}$ of training trajectories, $V(s)$ from TD learning
2: **for** each $\tau_i = \{s_0, a_0, \hat{G}_0, s_1, a_1, \widehat{G}_1, ...\}$ in $\mathcal{D}$ **do**
3:     Compute $\pi_\theta = (a_t | s_{t-k:t}, a_{t-k:t-1}, \widehat{G}_{t-k:t})$
4:     Calculate $L_\theta(\tau) = - \sum_t \log \pi_\theta(a_t | s_{t-k:t}, a_{t-k:t-1}, \widehat{G}_{t-k:t})$
5:     Backpropagate gradients w.r.t $L_\theta(\tau)$ to update model parameters $\theta$
6: **end for**
    {Evaluation of D2T2}
7: **Input:** Initial state $s_0$, behavior cloning transformer model $\pi_\zeta$
8: **for** $t = 0$ **to** $T - 1$ **do**
9:     $\widehat{G}_t = \bar{g}_\zeta(s_{t-k:t})$ (optional: $\widehat{G}_t \sim q_\psi(\cdot | \bar{g}_\zeta)$)
10:     Acquire action from $\pi_\theta = (a_t | s_{t-k:t}, a_{t-k:t-1}, \widehat{G}_{t-k:t})$
11:     Receive next state from environment
12: **end for**

---

## B  METHOD SUPPLEMENTARY

**(Optional).**  As discussed in Section 3.1, given that the value function $V(\cdot)$ is considered sub-optimal, its approximation errors can be propagated into the downstream supervised model $\bar{g}_\zeta(\cdot)$ trained following equation 5. As a result, directly using $\bar{g}_\zeta(\cdot)$ as the guidance signal for DT could be problematic, as the errors from the previous two steps can all be propagated into DT, in addition to the supervised learning error from the training of DT itself. To resolve this, we leverage variational inference (Kingma & Welling, 2013) to concentrate the learned knowledge into a compact and expressive latent space; as existing works have found that the latent space can facilitate detecting similar objects by devising a mapping over the $L_2$ distances in the latent space, such as stochastic neighbor embedding Van der Maaten & Hinton (2008). Consequently, we train a variational auto-encoder (VAE) to formulate such a space over $\bar{g}_\zeta(\cdot)$. Specifically, given a set $\bar{\mathcal{G}} = \bigcup_{\tau_i \in \mathcal{D}} \bar{G}(\tau_i)$, where $\bar{G}(\tau_i) = \{\hat{G}_0, \hat{G}_1, \ldots, \hat{G}_T\}$, where $\hat{G}_t = \bar{g}_\zeta(s_{t-k:t})$ with $s_t \in \tau_i, t \in [0, T]$, one can train a VAE to reconstruct $\bar{\mathcal{G}}$ by maximizing the evidence lower bound (ELBO), *i.e.*,

$$\max_{\phi, \psi} \frac{1}{N_{\bar{\mathcal{G}}}} \sum_{\bar{g} \in \bar{\mathcal{G}}} \Big[ \log p_\phi(\bar{g} | z) - KL(p(z) || q_\psi(z | \bar{g}_\zeta)) \Big]; \tag{7}$$

here, $p(z)$ is the latent prior following a normal Gaussian distribution, $q_\psi(z | \bar{g}_\zeta)$ is the approximated posterior that encodes $\bar{g}_\zeta$ to the latent space, and $p_\phi(\bar{g} | z)$ is the decoder. Following from Kingma & Welling (2013), both $\phi$ and $\psi$ are neural networks that output the mean and diagonal variance of $p_\phi$ and $q_\psi$ which both follow Gaussian distributions. Finally, we can leverage $q_\psi$ to map each $\bar{g}_\zeta \in \bar{\mathcal{G}}$ to its latent representation which we use as the SG, $\hat{G}_t \sim q_\psi(\cdot | \bar{g}_\zeta)$. As $\bar{g}_\zeta$ have already captured the state value information from future steps, and considered that such knowledge has been encapsulated into the latent space, we envision SG to be able to capture long-term goals for DT to achieve, as well as provide guidance for DT during evaluation.

The training and testing stage of D2T2 are described in Section 3.2 and summarized in Algorithm 2.

## C  EXPERIMENTAL DETAILS

We run all our experiments on Nvidia RTX A5000 with 24GB RAM and each experiment setting is averaged over 10 trials with different random seeds.

### C.1  TAILGATE DRIVING

In this stochastic problem, we collect around $100k$ samples and train the RL policies with $300k$ total timesteps. Our environment setting and baseline comparison are partly from (Villaflor et al., 2022). However, note that Villaflor et al. (2022) only considers one stop sign and does not investigate the data quality with crash rate. We list 3 different datasets in 4. To decide a better hyper-parameter

settings, we investigate the hyper-parameters and their corresponding tables with stop sign $= 50m$ as follows:

1. The discount factor $\gamma$ of the mapping function in the range of [0.7, 0.95]. Note that the $\gamma$ here is for steering guidance function mapping instead of the inputs for DT. In DT, the discount factor is 1.0.
2. The number of layers of the transformer model in the list of [2, 3, 4, 5]
3. The number of self-attention blocks in the list of [4, 8, 16]
4. The value of the context in the list of [3, 4, ..., 10]

Table 5: The discount factor $\gamma$ of the mapping function in the range of $[0.7, 0.95]$

| Metric | 0.70 | 0.75 | 0.80 | 0.85 | 0.90 | 0.95 |
|---|---|---|---|---|---|---|
| return | 0.75±0.03 | 0.83±0.025 | 0.82±0.01 | **0.86±0.03** | 0.83±0.045 | 0.79±0.02 |

Table 6: The number of layers of the transformer model in the list of $[2, 3, 4, 5]$

| Metric | 2 | 3 | 4 | 5 |
|---|---|---|---|---|
| return | 0.83±0.035 | 0.84±0.019 | **0.86±0.03** | 0.84±0.017 |

Table 7: The number of self-attention blocks of the transformer model in the list of $[4, 8, 16]$

| Metric | 4 | 8 | 16 |
|---|---|---|---|
| return | 0.81±0.06 | **0.86±0.03** | 0.85±0.02 |

Table 8: The value of the context $k$ in the list of $[3, 4, ..., 10]$

| Metric | 3 | 4 | 5 | 6 | 7 | 8 | 9 | 10 |
|---|---|---|---|---|---|---|---|---|
| return | 0.76±0.06 | 0.82±0.03 | **0.86±0.03** | 0.83±0.055 | 0.85±0.014 | 0.81±0.04 | 0.83±0.027 | 0.83±0.045 |

We conclude that the discount factor of mapping function and context length are relatively critical hyper-parameters in D2T2 for tailgate driving tasks. We list the additional hyper-parameters of transformer we choose for BC, DT, and D2T2 in Table 9, and TT in Table 10.

## C.2 STOCHASTIC FROZENLAKE

FrozenLake environment is a standard toy text environment in Open AI gym with discrete action and state spaces in dimensions of 4 and 16 respectively. In Figure 3, we show that the agent starts with **Start** point to achieve **Goal** along with **Frozen** while avoiding **Hole**. To modify it as a stochastic environment, the setup can be referred to in Section 4 and Yang et al. (2023).

Our D2T2 integrates vanilla DT with Temporal Difference learning. For stochastic FrozenLake task, we follow the hyper-parameters and codebase used in (Yang et al., 2023) with the well-trained Q-learning model under the hyper-parameters in (Kostrikov et al., 2022). We add extra code to replace RTG with our learned value functions after transformer-based behavior cloning on steering guidance.

## C.3 CARLA BENCHMARK

We evaluate our method on the CARLA NoCrash (Dosovitskiy et al., 2017; Codevilla et al., 2019) benchmark whose goal is to navigate in a suburban town to a desired goal waypoint from a predetermined start waypoint without crash, considering safety as a priority Wen et al. (2020); Hsu et al. (2022) in tasks such as autonomous driving. The benchmark consists of 2 towns *Town01* and *Town02*,

Table 9: The hyper-parameter used for BC, DT, and D2T2 for tailgate driving

| Hyper-parameters | Values |
|---|---|
| Discount factor | 1.0 |
| No of layers | 4 |
| No of heads | 8 |
| No of embed | 16 |
| action weight | 5 |
| reward weight | 1 |
| value weight | 1 |
| Batch Size | 256 |
| Learning Rate | 0.0001 |

Table 10: The hyper-parameter used for TT for tailgate driving

| Hyper-parameters | Values |
|---|---|
| Discount factor | 0.99 |
| No of layers | 4 |
| No of heads | 8 |
| No of embed | 16 |
| action weight | 5 |
| reward weight | 1 |
| value weight | 1 |
| Batch Size | 256 |
| Learning Rate | 0.0006 |

each with 25 different routes. We train our method D2T2 as well as all the baselines on the Town01 dataset and then evaluate them in the unseen Town02 routes.

NoCrash task is relatively more realistic and difficult compared with tailgate driving task. The state space includes various parameters such as relative heading error, distance from the target lane center, ego vehicle speed, relative distance to the leading vehicle (or max sensing range if none), speed of the leading vehicle (or max speed if none), and distance to the upcoming red light (or max sensing range if none). The reward consists of an increase with higher speed, a slight penalty for land deviations, and a huge penalty for crashes or traffic infractions. The trajectory terminates when the car crashes, incurs an infraction, times out, or reaches its destination.

Next, we evaluate our method on a modified version of the CARLA Leaderboard benchmark following the setting in Villaflor et al. (2022). Compared with NoCrash Benchmark, the policy is learned to change lanes in urban or highway situations. Since the goal is not just following the leading vehicle, 8 additional variables are included in the state space, providing the distance and speed of surrounding vehicles in each of the 4 diagonal directions.

Our D2T2 integrates vanilla DT with Temporal Difference learning. We keep the general Transformer hyper-parameters consistent as Villaflor et al. (2022) with 4 layers of self-attention blocks with 8 heads and 128 as an embedding size. We add extra code to replace RTG with our learned value functions after transformer-based behavior cloning on steering guidance.

## C.4 D4RL SUITES

Although our D2T2 aims to address stochastic problems, we also provide comprehensive studies on near-deterministic tasks in Gym-MuJoCo suite (Table 3), AntMaze-v2 suite (Table 11), and FrankaKitchen-v0 suite (Table 12. We do conduct experiments for Gym-MuJoCo tasks on D2T2 without learning latent representation. However, we do not list it in Table 3 because they are relatively less challenging compared with AntMaze and FrankaKitchen tasks so that the performance difference between with and without VAE is not significant.

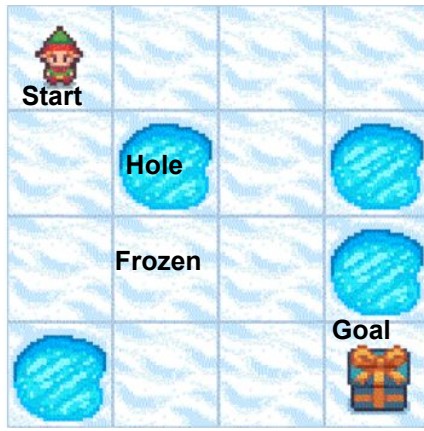

Figure 3: Visualization of the stochastic FrozenLake task.

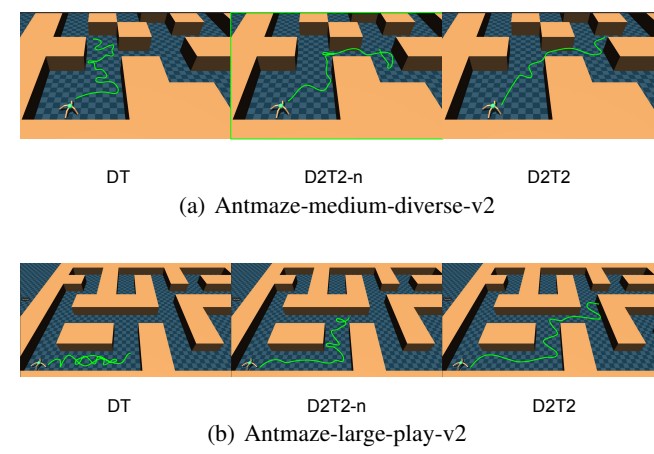

DT        D2T2-n        D2T2

(a) Antmaze-medium-diverse-v2

DT        D2T2-n        D2T2

(b) Antmaze-large-play-v2

Figure 4: Evaluation among DT, D2T2-$n$, and D2T2. The goal of both certain environments is at the top right corner and the path is recorded in green. (a): Antmaze-medium-diverse-v2. (b): Antmaze-large-play-v2

Table 11: Averaged normalized scores on AntMaze-v2 suite. We use the results reported from the RvS paper for BC and RvS-G, and the results from the IQL paper for CQL and IQL. For SPLT and CL, we use the results reported from their own papers respectively. We report the mean and for our method over 10 seeds. Note that D2T2-$n$ indicates learning D2T2 without latent representation.

| Environment | BC | RvS-G | DT | CL | SPLT | CQL | IQL | D2T2-$n$ | D2T2 |
|---|---|---|---|---|---|---|---|---|---|
| umaze-v2 | 54.6 | 65.4 | 65.6 | 79.8±1.4 | 70.5 | 74.0 | 87.5 | 65.2±2.7 | **87.9±0.7** |
| umaze-diverse-v2 | 45.6 | 60.9 | 51.2 | 77.6±2.8 | 40.2 | **84.0** | 62.2 | 52.7±1.1 | 69.9±1.9 |
| **average-umaze-v2** | 50.1 | 63.2 | 58.4 | 78.7 | 55.4 | **79.0** | 74.9 | 59.0 | 78.9 |
| medium-play-v2 | 0.0 | 58.1 | 1.0 | 72.6±2.9 | 25.0 | 61.2 | 71.2 | 42.5±2.0 | **72.7±2.3** |
| medium-diverse-v2 | 0.0 | 67.3 | 0.6 | 71.5±1.3 | 15.3 | 53.7 | 70.0 | 40.6±0.4 | **72.6±1.8** |
| **average-medium-v2** | 0.0 | 62.7 | 0.8 | 72.1 | 20.2 | 57.5 | 70.6 | 41.6 | **72.7** |
| large-play-v2 | 0.0 | 32.4 | 0.0 | **48.6±4.4** | 2.5 | 15.8 | 39.6 | 18.2±1.6 | 44.1±3.2 |
| large-diverse-v2 | 0.0 | 36.9 | 0.2 | **54.1±5.5** | 2.5 | 14.9 | 47.5 | 27.0±2.1 | 50.8±4.6 |
| **average-large-v2** | 0.0 | 34.7 | 0.1 | **51.4** | 2.5 | 15.4 | 43.6 | 22.6 | 49.7 |
| **average-Antmaze-v2** | 16.7 | 53.5 | 19.8 | **67.4** | 26.0 | 50.6 | 63.0 | 41.0 | 67.1 |

Table 12: Averaged normalized scores on FrankaKitchen-v0 suite. We use the results reported from the RvS paper for BC, RvS-G and RvS-R, and the results from the IQL paper for CQL and IQL. We report the mean and for our method over 10 seeds. Note that D2T2-$n$ indicates learning D2T2 without latent representation. Since DT does not report its official results, we do not include it here for fair comparison.

| Environment | BC | RvS-G | RvS-R | CQL | IQL | D2T2-$n$ | D2T2 |
|---|---|---|---|---|---|---|---|
| kitchen-complete-v0 | **65.0** | 50.2 | 1.5 | 43.8. | 62.5 | 61.1±0.9 | 62.3±1.4 |
| kitchen-partial-v0 | 38.0 | **51.4** | 0.5 | 49.8 | 46.3 | 43.1±2.0 | 47.8±2.5 |
| kitchen-mixed-v0 | 51.5 | **60.3** | 1.1 | 51.0 | 51.0 | 51.2±1.3 | 52.2±1.9 |
| **average-Kitchen-v0** | 51.5 | 54.0 | 1.0 | 48.2 | 53.3 | 51.8 | **54.1** |

On the other hand, The performance of D2T2 in AntMaze and FrankaKitchen tasks will be influenced by the dimension of the state/steering guidance space and the difficulty level of the data experience for imitation in high-dimensional problems. Therefore, we learn steering guidance over latent representation and report different versions of D2T2 for ablation study, where D2T2-$n$ denotes the algorithm learning without latent representation.

FrankaKitchen tasks contain the experience that is easier to imitate Fu et al. (2020), so we observe that D2T2-$n$ (without latent steering guidance) is still competitive in Table 12 but relatively poorly on the AntMaze tasks in Table 11. With latent steering guidance as input to D2T2, it performs comparably to the best-performing prior method, CL, on AntMaze tasks while CL is only developed for goal-conditioned offline RL. Specifically, in Figure 4 we show that D2T2-$n$ is already able to get closer to the goal compared with vanilla DT while D2T2 can even extract information, resulting in better performance.

Our D2T2 integrates vanilla DT with Temporal Difference learning. For D4RL suites, we follow the hyper-parameters and codebase used in (Chen et al., 2021) with the well-trained Q-learning model under the hyper-parameters in (Kostrikov et al., 2022). We add extra code to replace RTG with our learned value functions after transformer-based behavior cloning on steering guidance.

