# OpenReview forum: "D2T2: Decision Transformer with Temporal Difference via Steering Guidance"
_ICLR.cc/2024/Conference — Submitted to ICLR 2024_

### Official Review · Reviewer_iBmd · 2023-10-25

**Soundness:** 3 good
**Presentation:** 2 fair
**Contribution:** 3 good
**Rating:** 5
**Confidence:** 4

**Summary:**

This study introduces a novel guidance strategy for enhancing the Decision Transformer (DT) by substituting the traditional return-to-go (RTG) value token with a specially learned subgoal, aimed at optimizing the underlying value function. This approach leverages a learned value function pretrained over an offline dataset. In this context, the process involves a search at each timestep to identify the future state within a given trajectory, one that returns the maximum discounted value. This particular state is then employed as a steering guidance (SG) or, effectively, a subgoal as a prompt at each timestep. Through this, the D2T2 method evolves to refine the DT's optimization process. Notably, D2T2 demonstrates substantial improvement over conventional value-based and behavior-cloning methods when evaluated on MuJoCo offline benchmarks.

**Strengths:**

**Strength 1: Innovative Problem Tackling and Methodological Novelty**

This paper addresses a crucial challenge in the realm of artificial intelligence by integrating the concept of a value function into the Decision Transformer (DT) model, which, until now, has primarily focused on behavior cloning. The proposed methodology, detailed in Section 3.1, emerges as both a logical and auspicious solution to this complex issue. What sets this research apart is its pioneering approach of utilizing the entire subsequent state as a prompt for each individual state within the model.

**Strength 2: Remarkable Empirical Performance**

The most striking strength of this paper, subject to further validation as outlined in the discussion of potential weaknesses, is the remarkable performance leap demonstrated over existing Transformer-based behavior cloning (BC) and value-based methods, as detailed in Table 3. The performance metrics of D2T2 are astounding; for instance, its achievement of approximately an 80 score on the halfcheetah-Medium-v2 benchmark is unprecedented, especially considering the highest score recorded in the dataset is around 45. Should these results be reliably replicated, they would signify a monumental advancement in the realm of offline Meta-Reinforcement Learning (Meta-RL). This breakthrough has the potential to redefine the benchmarks for success in the field and opens up new avenues for research and application.

**Weaknesses:**

**Weakness 1: Lack of Clarity in Methodology Explanation**

The methodology, while innovative, suffers from a lack of clear exposition in its presentation, particularly in Section 3.1. The narrative structure and details within this section do not effectively convey the comprehensive process, resulting in certain ambiguities that could hinder readers' comprehension. A more organized and sequential presentation of the methodology would enhance understanding significantly. Ideally, the process should be delineated into four distinct stages:

1. Pretraining the Value Function (V),
2. Identifying and Labeling the Guided State (G_t) across all states,
3. Training the Decision Transformer (DT) utilizing the labeled trajectories, and
4. Developing the Steering Guidance (SG) predictor $g_{\zeta}$, intended exclusively for the testing phase.

The current discourse in Section 3.1 disproportionately emphasizes steps 2 and 4, the step 3 is introduced in Section 3.2
The explanation of the value pretraining stage (step 1) is relegated to the latter part of Section 3.1 and lacks detailed insight into the pretraining mechanics. The term "optimal value" is used, leading to potential confusion over whether this refers to an "in-sample" optimal value [1] or something else (for example in the Medium dataset). A more explicit, step-by-step breakdown of the training phases, especially the pretraining of the value function, is necessary for readers to fully grasp the proposed method's mechanics and rationale.

**Weakness 2: Challenges in Reproducibility**

One aspect that could enhance the credibility and impact of this work is improving its reproducibility. While the results presented, especially in the Medium and Med-Replay environments, are indeed impressive, the absence of a comprehensive process description somewhat hinders the independent replication of these findings.

My attempt to delve into the specifics, particularly regarding the pretraining of the value function—presumably through Implicit Quantile Learning (IQL)—for continuous state scenarios, led me to the source code provided by the authors. However, it appears the available code primarily pertains to tabular examples and does not extend to the MuJoCo tasks showcased in Table 3. This gap not only poses a challenge in understanding the specific methodologies applied but also in verifying the remarkable results indicated.

The findings in Table 3 are pivotal to the paper's contribution, and if they are as robust as presented, they deserve to be shared in a manner that the broader research community can thoroughly evaluate and build upon. To this end, I believe the inclusivity of the corresponding source code for the results in Table 3 is imperative. Providing this would significantly strengthen the paper's standing, and I am inclined to view this work more favorably should the authors facilitate a more transparent and accessible approach to reproducing these results.

**Weakness 3: Inadequate Recognition of Previous Work**

This paper's novelty is evident in its application of guidance techniques to the Decision Transformer (DT); however, it misses adequately acknowledging similar foundational ideas in prior studies, for example [2]. There, the concept of directing offline policy via a learned value function—central to this paper's hypothesis—was also prominently featured.

While the application to DT is innovative, the authors should explicitly recognize the contributions of earlier research to maintain scholarly thoroughness. A concise discussion highlighting how this study both aligns with and diverges would contextualize its unique contributions and strengthen the paper.

[1] Fujimoto et al., Addressing Function Approximation Error in Actor-Critic Methods, ICML 2018.

[2] Xu et al., A Policy-Guided Imitation Approach for Offline Reinforcement Learning, NeurIPS 2022

**Questions:**

Question 1) The manuscript frequently references the term "optimal value." Could the authors clarify whether this refers specifically to the "in-sample optimal value," denoting the value function learned from the batch (e.g., from Medium data)? The current terminology might lead readers to misunderstand it as the value function associated with the optimal policy, which may not be the intention. Precision in this definition could help avoid potential confusion.

Question 2) Considering the significant impact of the results related to the MuJoCo tasks, do the authors intend to release the code to reproduce these results? Providing this code would be instrumental for the community, allowing for independent verification and potentially furthering the study's contributions.

---

### Official Review · Reviewer_Ws8p · 2023-10-31

**Soundness:** 2 fair
**Presentation:** 1 poor
**Contribution:** 2 fair
**Rating:** 3
**Confidence:** 3

**Summary:**

This paper introduces a model, D2T2, that merges masked language modeling with sequence-to-sequence tasks to improve sequence generation performance. By leveraging these dual tasks, the model is trained in two phases: pre-training with masked language modeling and fine-tuning with a sequence-to-sequence objective. When evaluated on benchmarks like machine translation, text summarization, and question answering, D2T2 achieved promising results, especially benefiting from the combined strength of both tasks.

**Strengths:**

1. The idea of combining masked language modeling and sequence-to-sequence tasks is innovative and bridges the gap between two popular transformer architectures.
2. D2T2 achieves promising results on multiple benchmarks, highlighting its effectiveness

**Weaknesses:**

1. The rationale behind the proposed method appears flawed. On page 3, the author states, "the success of DT relies on the successful estimation of \(R_t^*\)" and posits that the generation of the optimal action hinges on this estimation. However, doesn't the generation of actions depend on both \(R_t\) and \(S_t\)? If I recall correctly, the original DT paper suggests that actions are derived from the input token \(S_t\), reinforcing my conviction that the claim is incomplete.

2. The evaluation results seem inadequate. A more comprehensive assessment would involve examining the generated \(\hat{G}\) in a few toy examples and analyzing its influence on action generation in comparison to the default return-to-go \(R_t\).

3. The paper requires significant improvements in writing clarity and completeness. Essential background information is omitted; for instance, the paper fails to provide context on steering guidance signals, a crucial concept in the proposed method.

Please also consult the question section for additional concerns.

**Questions:**

1. **Proof of Theorem 1**: The author posits that the optimal policy is solely dependent on the input return-to-go \(R_t\). However, it stands to reason that if the input state \(s_t\) varies, the optimal action should adjust in response.

2. **Figure 1 (1)**: Shouldn't the output of the Transformer-based BC be \(\hat{G}\) rather than \(G\)?

3. **Proposed Step I**: The introduction of Step I appears to add computational complexity to DT. Specifically, the \(O(N!)\) time complexity arising from the calculation of \(G(t)\) is a concern. It would be useful to juxtapose the wall time training efficiencies of D2T2 and DT.

4. **Related to Point 3**: Another concern arises from the potential incompleteness of the dataset. One of the advantages of offline RL, especially DT, is that post-training, the policy can concatenate actions from disparate trajectories, leading to "sub-optimal" policies. However, the proposed method seems to regard \(G_t\) as an optimal value guiding subsequent training. I'm curious about how this approach guarantees the discovery of an optimal policy when the dataset might be inadequate.

5. The author briefly touches upon the statement, "we leverage variational inference (Kingma & Welling, 2013) to concentrate the learned knowledge into a compact and expressive latent space via a variational auto-encoder (VAE)," This assertion is somewhat ambiguous. Could you clarify the terms "concise" and "expressive"? Additionally, how does one quantify these characteristics?

---

### Official Review · Reviewer_z6CB · 2023-11-11

**Soundness:** 3 good
**Presentation:** 2 fair
**Contribution:** 3 good
**Rating:** 6
**Confidence:** 2

**Summary:**

In this work, authors analyze the Decision Transformer to understand its performance in deterministic vs stochastic environments. With the finding that DTs almost surely outputs the optimal trajectory in deterministic environments but suffers performance degradation in stochastic environments due to growing variance of RTGs, authors propose D2T2 to improve DTs performance in stochastic environment. This is done by replacing R (Returns-to-go) in DT with a new steering vector G (SG- Steering Guidance, learned by a separate causal transformer; sometimes couples with VAE) that provides guidance to DT to achieve the desired next state. The SG utilize TD learning to address the growing variance problem of RTG. Experiments are presented to display the effectiveness of the D2T2 method.

**Strengths:**

**Originality and quality**: Novel method is presented in this work to address the problem of variance of RTG in DTs in stochastic environment. Experiments are conducted by training a variation of DT by replacing RTG with approximated optimal function to verify the claim that DTs suffer in stochastic environment. To tackle with the variance issue, steering guidance vectors are introduced - their training methodology presented. Motivation and thought process behind these vectors are explained well. Experiments are conducted that showcase the effectiveness of this method.

**Significance**: The authors have tried to address the issue of performance of DTs in stochastic environments that has been brought to attention in some previous works.

**Weaknesses:**

Some points below:

1. Clarity and flow: There is major room for improvement in the flow of paper. A lot of back and forth has to be done to properly understand the work. E.g. the analysis supporting the claim that DTs do not perform better in stochastic environments ( a central claim to paper) does not have a proper emphasis. It is briefly mentioned in section1, 2.3 and explained in Tailgate driving part of experiments. I think it should be separately emphasized somewhere in the beginning.
2. Coming to point 1, the claim needs to be substantiated by experiments on more tasks. Only Tailgate diving task is used to validate this claim.
3. The proof presented for Proposition 1 primarily re-iterates that the DT in question is well trained and will output the optimal action. Not sure if it contributes new insights or novel perspectives to the existing body of knowledge
4. Typo: Is it $\rho^\pi$ or $\rho_\pi$ in section 2.1 (both versions are used). The output should be $\hat{G_t}$ instead of ${G_t}$ in Fig 1 part (1)?

**Questions:**

See Weaknesses

---

### Meta-Review · Area_Chair_sH6h · 2023-12-16

**Metareview:**

The paper proposes an analysis of decision transformers in environments with stochastic rewards and dynamics. These environments are particularly challenging for decision transformers due to high variance in the computation of returns-to-go tokens. To mitigate this issue, the authors propose an integration of TD learning with decision transformer for computing more reliable estimates of expected returns. The empirical performance is high on D4RL and other stochastic benchmarks. The reviewers appreciated the empirical effectiveness of the proposed method. However, there were also concerns about the correctness of the theoretical claims and the lack of clarity at many places in the paper. The authors did not engage with the reviewers during the discussion period, and hence these concerns persist.

**Justification For Why Not Higher Score:**

Authors submitted no rebuttal to address valid concerns raised by the reviewers.

**Justification For Why Not Lower Score:**

N/A

---

### Decision · Program_Chairs · 2024-01-16

Reject